# Reconceiving the hippocampal map as a topological template

**Yuri Dabaghian[1,2]\*, Vicky L Brandt[1,2], Loren M Frank[3,4]**

[1]The Jan and Dan Duncan Neurological Research Institute at Texas Children's Hospital, Houston, United States; [2]Baylor College of Medicine, Houston, United States; [3]Sloan-Swartz Center for Theoretical Neurobiology, W.M. Keck Center for Integrative Neuroscience, University of California, San Francisco, San Francisco, United States; [4]Department of Physiology, University of California, San Francisco, San Francisco, United States

**Abstract** The role of the hippocampus in spatial cognition is incontrovertible yet controversial. Place cells, initially thought to be location-specifiers, turn out to respond promiscuously to a wide range of stimuli. Here we test the idea, which we have recently demonstrated in a computational model, that the hippocampal place cells may ultimately be interested in a space's topological qualities (its connectivity) more than its geometry (distances and angles); such higher-order functioning would be more consistent with other known hippocampal functions. We recorded place cell activity in rats exploring morphing linear tracks that allowed us to dissociate the geometry of the track from its topology. The resulting place fields preserved the relative sequence of places visited along the track but did not vary with the metrical features of the track or the direction of the rat's movement. These results suggest a reinterpretation of previous studies and new directions for future experiments.

**\*For correspondence:**
dabaghia@bcm.edu

**Competing interests:** The authors declare that no competing interests exist.

**Reviewing editor**: Howard Eichenbaum, Boston University, United States

## Introduction

When O'Keefe and Dostrovsky discovered that the firing of hippocampal place cells correlates strongly with the position of the rat with respect to discrete locations within the environment (the cell's place field) (**O'Keefe and Dostrovsky, 1971**), they launched decades of intensive research into how place cells contribute to an internal map of the environment (**Best et al., 2001**). That there is a considerable gulf between place cell firing and a cognitive spatial map, however, has become only clearer with time. Place cells appear to respond to a perplexing array of stimuli, from visual cues to head direction, goal planning, color changes, shape changes, and olfactory, vestibular and kinesthetic inputs, and discrepancies between the expected and actual location of a target (**Harnad, 1994**; **Frank et al., 2000**; **Wood et al., 2000**); recent work has shown that both spatial and nonspatial cell types in the entorhinal cortex provide myriad inputs to the hippocampus (**Ideker et al., 2011**). The fascination with place cell promiscuity, however, has tended to distract from the fact that cognition is the work not of individual neurons but of large ensembles of cells (**Ludvig, 1999**; **Fenton et al., 2008**). The nature of the information embedded in the place cell ensemble code and transmitted to downstream neurons has been largely ignored, along with the consequences for the type of spatial properties that might form the basis for the cognitive map.

Whatever the identity of these downstream neurons, they clearly have no direct access to the physical environment or to the place fields mapped by experimentalists. The only information they receive is contained in the temporal pattern of the spike trains of the thousands of place cells that are active in a given environment. What sort of information about a space might be constructed from such signals? The dominant assumption within the field of neuroscience has been that the hippocampal spatial

**eLife digest** The hippocampus is one of the most easily recognizable structures in the brain owing to its characteristic seahorse-like shape. Brain imaging studies in the 1990s famously showed the hippocampus to be larger in London taxi drivers than in other people, suggesting that it plays a role in spatial navigation. This was consistent with previous findings in rodents, which had shown that the hippocampus is active when animals find their way through mazes.

Electrode recordings have revealed that whenever an animal is in a specific location of a particular environment (for example, in the back left-hand corner of a small room with white walls) one or a small number of cells within the hippocampus will fire to encode that location. When the animal moves to a new location within the same environment, other cells will fire to encode the new location. In this way, the population of cells—which are known as place cells—can together construct a virtual 'map' of the environment.

It is generally assumed that this hippocampal map represents space in terms of the absolute distances and angles between locations, rather like a street map. However, this type of geometric map appears inconsistent with the results of certain experiments. Dabaghian et al. proposed instead that the hippocampal map is based on topology, or the relative order of locations and the connections between them, rather like a subway map. Subsequently, computer models demonstrated that virtual simulations of place cells could effectively 'learn' the topological features of different environments.

Now, Dabaghian et al. provide their own empirical data to support the existence of a hippocampal 'subway-style' map by recording the electrical impulses from place cells in the rat hippocampus as the animals ran through a U-shaped maze. The maze was constructed so that its arms could either be straight or folded into zigzags. Changing the maze in this way does not alter its topology because the relative order of its various components—such as the positions of food wells in the arms—is unchanged, but it does alter the maze's geometry. Notably, as rats ran through different conformations of the maze, the activity of the place cells in their hippocampi remained largely unchanged, consistent with a map based on topology rather than geometry.

By providing evidence that hippocampal maps have more in common with subway maps than street maps, the work of Dabaghian et al. offers an explanation for previously challenging results and provides a framework for further experiments into hippocampal memory function.

map—if there is one (*Eichenbaum et al., 1999*)—is an allothetic, Euclidian representation of the local environment that integrates detailed information about distances and angles arising through self-motion, head direction, and speed in addition to visual and other cues (*Wills et al., 2010*; *Erdem and Hasselmo, 2012*; *Chen et al., 2013*; *Rubin et al., 2014*). The compilation of such geometric data would seem to require considerable computational power for the spatial map to be formed quickly enough to be useful under normal conditions of animal navigation (*Wilson and McNaughton, 1993*; *Frank et al., 2004*). Geometry is not the only aspect of space that can form the basis of a map, however. From a practical standpoint, the topological qualities of a space—relations between locations such as continuity, enclosure, sequence (*Poincaré, 1895*; *Aleksandrov, 1965*)—are arguably even more important (*Poucet, 1993*). Yet scant attention has been paid to the possible role of topological features in the cognitive map, despite the fact that any movement through space is describable in topological terms (locations A and B must be somehow connected if movement between them is possible). This neglect is even more curious in light of Piaget's early work indicating that children first represent space topologically and only later learn to incorporate metrical details into their mental representations of space (*Poucet, 1993*; *O'Keefe and Nadel, 1978*). A few studies have shown that place cell firing is responsive to topological changes in the environment such as the placement or removal of a barrier across a previously learned route (*Poucet and Herrmann, 2001*; *Alvernhe et al., 2011*), but as noted above, place cells seem to be responsive to an enormous variety of inputs. Our interest ultimately is not whether place cells can respond to topological features of a space, but whether the information conveyed downstream by place cell spiking might encode topological information, which would offer considerable advantages over geometry in terms of computational speed and flexibility.

There are two compelling reasons to consider such a radical departure from the dominant assumptions about the spatial component of the cognitive map. First is the nature of place cell firing. If, in fact, place cell co-firing implies spatial overlap of place fields (*McNaughton et al., 1983*; *Brown et al., 1998*; *Zhang et al., 1998*; *Jensen and Lisman, 2000*; *Barbieri et al., 2001*, *2005*; *Guger et al., 2011*), then a map derived from place cell spiking will necessarily emphasize contiguities between locations as well as the temporal sequence in which they are experienced. (The reason we include temporal sequence along with more conventional spatial relationships is because movement through space takes place over time; sequence thus embodies connectivity.) This would seem to be more compatible with the broader role of the hippocampus in representing and storing experience. Second, how geometric changes would be transmitted downstream as metrical changes per se is unclear; precisely because place cells respond to so many inputs, a change in firing rate within a period of overlap might just as well reflect a change in the animal's speed or some other input as it would a change in distance. *Muller and Kubie (1987)* found that enlarging their open field set-up caused place fields to enlarge while maintaining the same shape and relative order (i.e., the connectivity of the environment remained invariant). Similar observations were made by *Gothard et al., 1996a* on a linear track, by *O'Keefe and Burgess (1996)* in rectangular environments, and by other studies in morphing environments (*Lever et al., 2002*; *Leutgeb et al., 2005*; *Touretzky et al., 2005*; *Wills et al., 2005*; *Colgin et al., 2010*). O'Keefe et al. (*O'Keefe and Burgess, 1996*) reasoned that the responses of the place fields could be described in terms of the geometric characteristics of the deforming environment, such as the altered distances to the walls of the enclosure. If we consider the information available to the downstream neurons that 'read' spiking coactivity, however, this explanation becomes a bit less plausible. As adjacent place fields enlarge in response to an expanded environment, the place cells will still be coactive, and the readout to the downstream neurons will be much the same. Thus, even if the place field layout stretches significantly, the corresponding geometrical information will not necessarily be captured by the downstream networks unless the change is quite large. Note that this phenomenon of place field stretching has a topological quality to it, since the fields stretch with the space but maintain their relative positions and order. In this context it is interesting to note that a more detailed analysis of place cell activity in a stretching *1D* environment (*Diba and Buzsaki, 2008*) showed that the temporal gaps between the spikes of the respective place cells remain fixed throughout the stretch, despite the change in the overall firing rate (rate remapping, *Dupret et al., 2010*). Mathematically speaking, this suggests that spatial information contained in the hippocampal map may be invariant with respect to a certain range of geometric transformations, so that the flexible arrangement of place fields does not represent specific locations on a Cartesian grid, but rather a coarser framework of spatiotemporal relationships.

Another advantage to the topological model is that it implies that questions of spatial learning might be amenable to recently developed approaches in the field of computational topology. The way place fields cover a space actually calls to mind a fundamental theorem in algebraic topology that holds that a given space, if covered by a sufficient number of discrete sub-regions, can be reconstructed by the pattern of overlaps between those sub-regions (a simplicial complex) (*Hatcher, 2002*). This theorem is key to the power of topological approaches to resolve otherwise overwhelmingly complex problems in high-dimensional data analysis (*Lum et al., 2013*), and it lends itself to the problem at hand. We recently developed a computational algorithm based on algebraic topology to simulate the behavior of a rat exploring several different environments (*Dabaghian et al., 2012*; *Arai et al., 2014*). We created a wide range of testable values for three parameters—place cell firing rate, number of place cells, and place field size—and showed that simulated groups of neurons accurately encode spatial information from topologically distinct environments (they 'learn' the space) when the parameter values happen to closely parallel biological values derived from animal experiments. Since we made no a priori assumptions about what values would work, this was a promising validation of our model. We subsequently included theta precession in the model and showed that theta oscillation affects the behavior of the neuronal ensemble in ways that augment its performance to enhance learning (*Arai et al., 2014*).

Still, there has been no experiment designed explicitly to examine how place cells respond to significant geometric changes when the topology is held invariant. Here we report such an investigation. Since the location-specific firing of place cells is thought to be produced by a path integration mechanism (accurate idiothetic counting of the elementary displacements and changes of orientation, speed and acceleration produced during movements [*McNaughton et al., 1996*; *Etienne and Jeffery,*

*2004*; *McNaughton et al., 2006*]), we used morphing linear tracks whose geometry limits the pool of possible confounding motions. If place cells are primarily concerned with encoding geometric information, then their firing activity should reflect the dynamic planar geometry of the track. If, on the other hand, place cells are primarily concerned with topological information, then variations of the track's configuration should have little effect on where the place cells fire.

## Results

We devised two elevated U-shaped tracks with a flexible construction that allowed the geometry of one arm to be significantly altered while the rat was on the other part of the track. On the first track, Track A, each arm of the U could bend at four joints to transition from a straight runway in the expanded configuration to a compressed zigzag (*Figure 1*; 'Materials and methods'). The dynamic configurations of the track were mapped against a stable set of spatial bins, which allowed us to dissociate two frames of reference: the effects produced on the CA1 place cells by changes in the track configuration can be described either in the reference frame associated with the room (the 2D planar reference frame) or with the track itself (the 1D linear reference frame). If path-integrating the 2D displacements, velocities and acceleration produces place fields that from an allocentric, Cartesian coordinate map of the space, the place fields should be relatively stable in the 2D plane, remaining in the same spatial bins despite changing track segments, and variable in the 1D frame of reference (See *Figure 1*, *Figure 1—figure Supplement 1*). In contrast, if the place cells attend to the sequence and connectivity of places experienced on the track irrespective of the track's geometry and the directions and angles of the rat's movements, then the place fields will be stable in the linear reference frame, but variable in the planar (2D) reference frame.

Three animals were exposed to the initially novel Track A. We recorded a total of 557 pyramidal cells in CA1 and analyzed the pattern of their activity in response to the expansions and contractions of the track in both reference frames ('Materials and methods') and compared the spatial firing rates

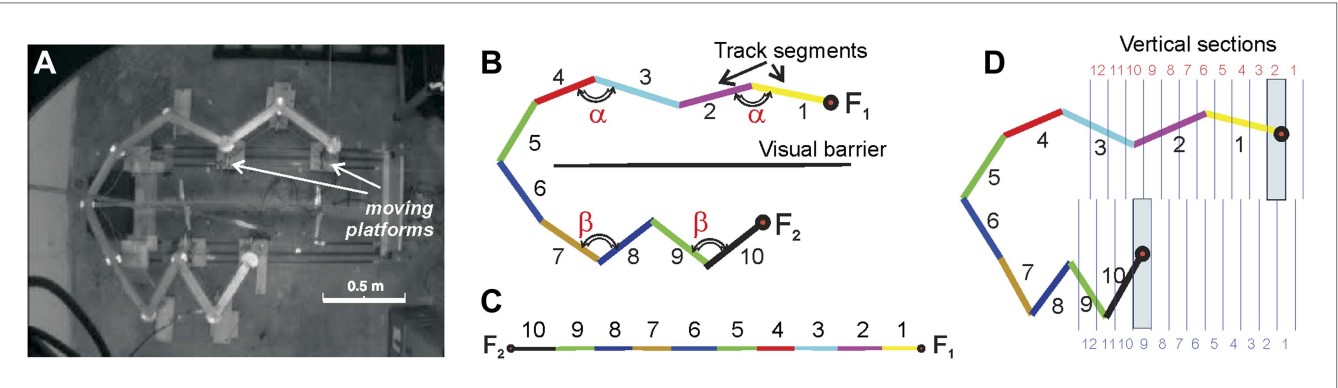

**Figure 1.** Compressible U/zigzag track (Track A). (**A**) Top-down view of the four-meter long track with both arms in a semi-contracted configuration. (**B**) A schematic representation of the track illustrating how the segments are numbered in the planar (**B**) and the linear (**C**) frames of reference. (Note that the track depiction in C is drawn at half the scale as in **B**. The track was designed so that the angle between the segments 1 and 2 and between segments 3 and 4 on the top arm remain the same whether the track arm is extended or compressed, that is, ∠(1,2) = ∠(3,4) = α. Similarly, the angles on the bottom arm remain equal, ∠(7,8) = ∠(9,10) = β. The magnitude of angle α is defined by the stretch of the top arm (by the position of the food well $F_1$), and angle β is defined by the position of the food well $F_2$. Thus, the geometric configuration of the track can be described either by the angles (α, β) or the positions of the food wells $F_1$ and $F_2$. (**D**) To more readily denote track arm configurations, we divided the space that can be occupied by the fully extended top and bottom arms into 12 vertical sections each. Each position of a track arm can be described by the numbered vertical section into which $F_1$ or $F_2$ (for the top and bottom arms, respectively) falls. In the depicted track configuration, $F_1$ falls into section 2 and $F_2$ falls into section 9. However, because we are interested in being able to denote changes in track segment positions in order to compute correlation coefficients, the top arm can be said to move from position *i* to position *j* (numbers in red), and the bottom arm from position *k* to position *l* (numbers in blue; see 'Materials and methods'). So the track configuration depicted here can be fully described by $F_1$ = 2 and $F_2$ = 9. Given movement of an arm from section *i* to *j* (or *k* to *l*), the larger the difference between the numbers, $d = |i − j|$ or $|k − l|$, the more geometrically dissimilar positions *i* and *j* or *k* and *l* are.

The following figure supplement is available for figure 1:

**Figure supplement 1**. Calculating correlation coefficients for place cells firing in different track configurations.

across track configurations. We found that, for the CA1 place cells active on the mobile sections of the track, spiking was stable in the linear but not in the *2D* reference frame. *Figure 2A,B* shows the spiking pattern of one representative neuron in two different track configurations, each of which is plotted in both the linear and planar reference frames. As evident in the overlay of the two plots (*Figure 2A+B*), the place cell's spiking was widely dispersed in the *2D* reference frame but localized in the linear frame. We quantified this tendency by computing the correlation between the firing rates in the planar and linear reference frames for all pairs of configurations. Because any two linearized track configurations are geometrically identical, whereas only a few parts of the *2D* track configurations overlap, we computed the correlations using only the spatial bins that were visited and where at least one spike was fired in both *1D* and *2D* configurations ('Materials and methods').

Since the track is divided into uniformly spaced vertical sections, the geometric dissimilarity between two configurations can be expressed simply as the difference between the section indices (See explanation in *Figure 1D*, 'Materials and methods'). Therefore, all the correlation coefficients with the same difference between the first and the second index, $C_{i,i+d}$, will describe the correlations between 'equally dissimilar' track configurations separated by $d$ bins and appear on the $d$-th subdiagonal of the correlation matrix. Their average value, $r_d = < C_{i,i+d} >$, thus characterizes the mean correlation for a given degree of dissimilarity, and the decay of $r_d$ as a function of $d$ characterizes the decay of correlations as a function of the difference between track geometries. The matrices in *Figure 2C* show that the correlations computed in the *2D* reference frame decayed rapidly as more geometrically dissimilar configurations were compared. In contrast, the linear correlations remained high even for the fully expanded configuration. The difference in the mean correlation is highly significant (p < 0.001, two sample $t$ tests) for each point (all values of $d$) on both curves (*Figure 2D*). The high values of the correlation coefficients for individual neurons in the linear case also show that the place fields were stable, despite the fact that active place cells could produce only a few spikes during the animal's single run through the corresponding place field in each track configuration, the effect of which was to create greater variability in our estimates of the firing rate and reduce the value of correlation coefficients (*Figure 2C*). To this end, we also examined cells that were active on the static portions of the track (*Figure 2E–F*) and found that they demonstrated uniformly high correlation values for both reference frames (*Figure 2G–H*). For these cells, the average values of the inter-configuration correlation coefficients in the *1D* and the *2D* case did not diverge: the place fields of these neurons were stable despite the motion of other track elements. This demonstrates that place fields on the moving segments shift in *2D* not because of some general instability but because they are, in fact, responding to changes in the configuration of the track.

The same place field stability in the linear frame is clearly evident in the ensemble averages for all place cells recorded over all days of recording for each animal (*Figure 3A–C*) and in the combined data across all cells from all animals (*Figure 3D,E*). As is clear from *Figure 3A–C*, the correlations were stable in the linear reference frame but decayed rapidly as a function of track configuration changes in the *2D* reference frame. At the same time, the correlations for cells active only on the static portion of the track showed no tendency to decay as a function of the distance between vertical sections as the configurations changed (compare *Figure 3D,E*). The place fields in the linear frame of reference were thus largely invariant with respect to changes in track geometry.

The observed stability of place cell activity patterns cannot be attributed to the animals' lack of awareness of the *2D* shape of the environment or to their habituation to the environment ('Materials and methods'). The animals were avidly curious about the space around the elevated track, and their deft performance on the constantly changing (if faintly glowing) track in a dark room suggests that the proprioceptive and idiothetic information required for such behavior was available to the animals because of the anatomical connections to the hippocampus (e.g., with head direction cells [*Calton et al., 2003*; *Navratilova et al., 2012*] and other inputs [*Sharp et al., 1995*; *Muir and Bilkey, 2003*; *Lu and Bilkey, 2010*]). Thus, it is likely that the recorded activity in area CA1 was stable despite the availability and the variability of these inputs.

It might be argued that the above results could be explained by a *1D* path integration mechanism that somehow excludes vestibular and some proprioceptive input so that the place cells respond simply to the distance that the animal traveled from/to the food wells (*Gothard et al., 1996a*; *Gothard et al., 1996b*). Such a reduced path integration mechanism is sometimes expanded ad hoc to include place cell binding to local features of the track, such as joints or segments (*Chen et al., 2013*; *O'Keefe and Burgess, 1996*). To evaluate these possibilities, we carried out an additional experiment in two

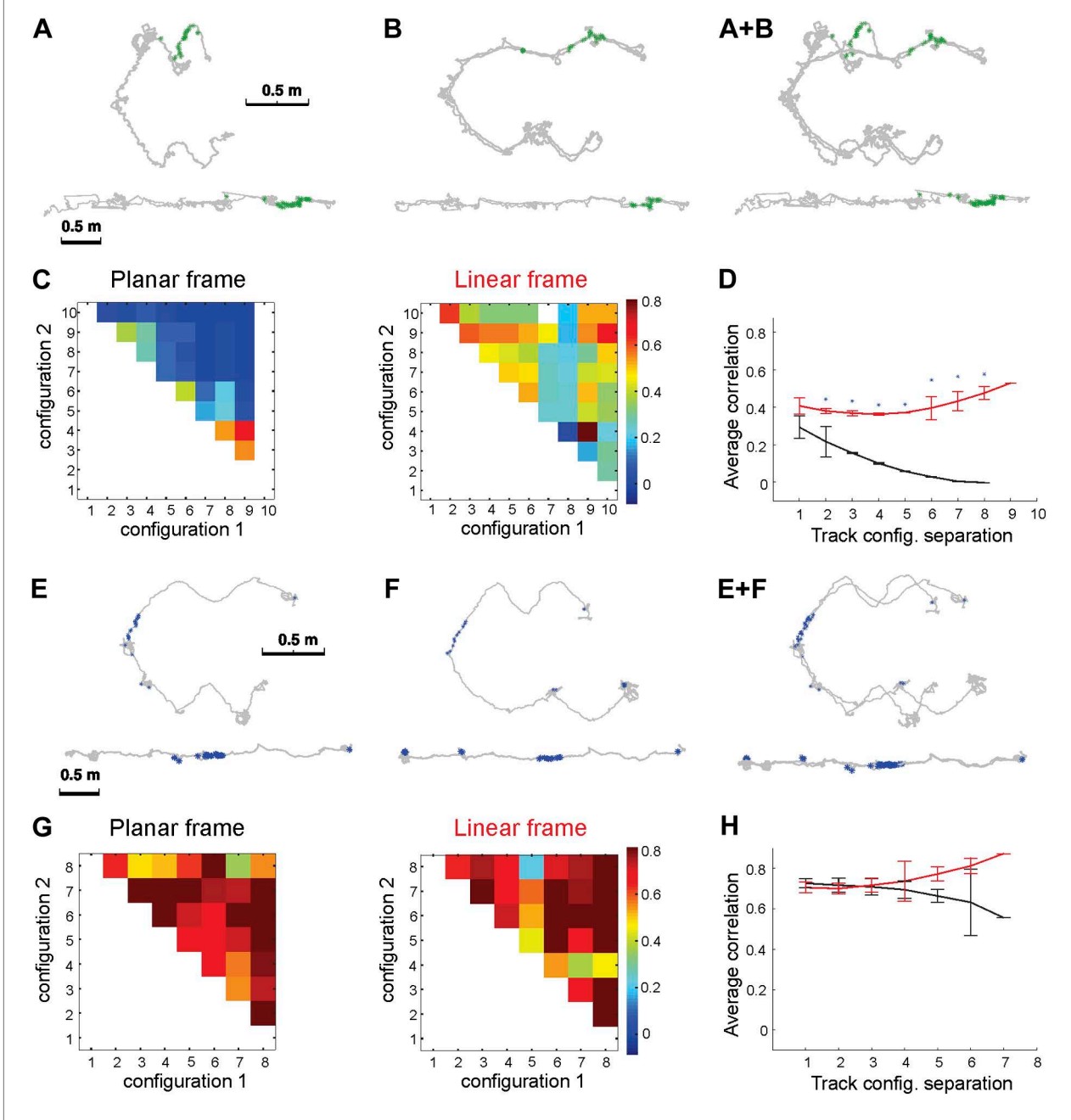

**Figure 2**. Place cell spiking across different configurations of Track A shows that place fields remain stable despite large geometric changes. Depicted are the firing patterns of two place cells: one cell that was active primarily on one dynamic segment of the track (**A–D**) and another place cell that was active on a static part of the track (**E–H**). (**A**) Each green dot represents a spike, which is shown in two reference frames: 2D (upper row) and linear (bottom row). The gray line represents the animal's path across all trials with these configurations. (**B**) Overlaying the 2D and linear reference frames shows that the spiking is distributed in 2D space but much more localized in the linearized reference frame. (**C**) The matrix of the correlations between the activity of this place cell over the moving sections of the track defined in planar (left) and in the linear (right) frame. The color of each square ($C_{ij}$) represents the mean correlation coefficient between two track configurations, configuration₁ and configuration₂. The correlations were clearly much higher in the linear frame. (**D**) The mean correlation coefficient values, $r_d = < C_{i,i+d} >$, averaged over all pairs of the track configurations separated by $d$ bins. The lines show whether the place cell firing was stable across configurations in the linear (1D, red) and the planar (2D, black) reference frames as a function of the difference $d$ between the two configurations; the black line shows clear decay of correlation between the configurations, meaning that place cell firing was not stable in 2D (See **Figure 1** for explanation of **D**). (**E–H**) The same plots for a different neuron (spikes shown in blue) that was active on the static track segments. The graph to the right shows a high correlation coefficient for both frames of reference, indicating that the place fields were stable. The asterisk above the SEM error bars indicates p < 0.001. A total of 557 neurons were recorded; the place cells depicted here are representative.

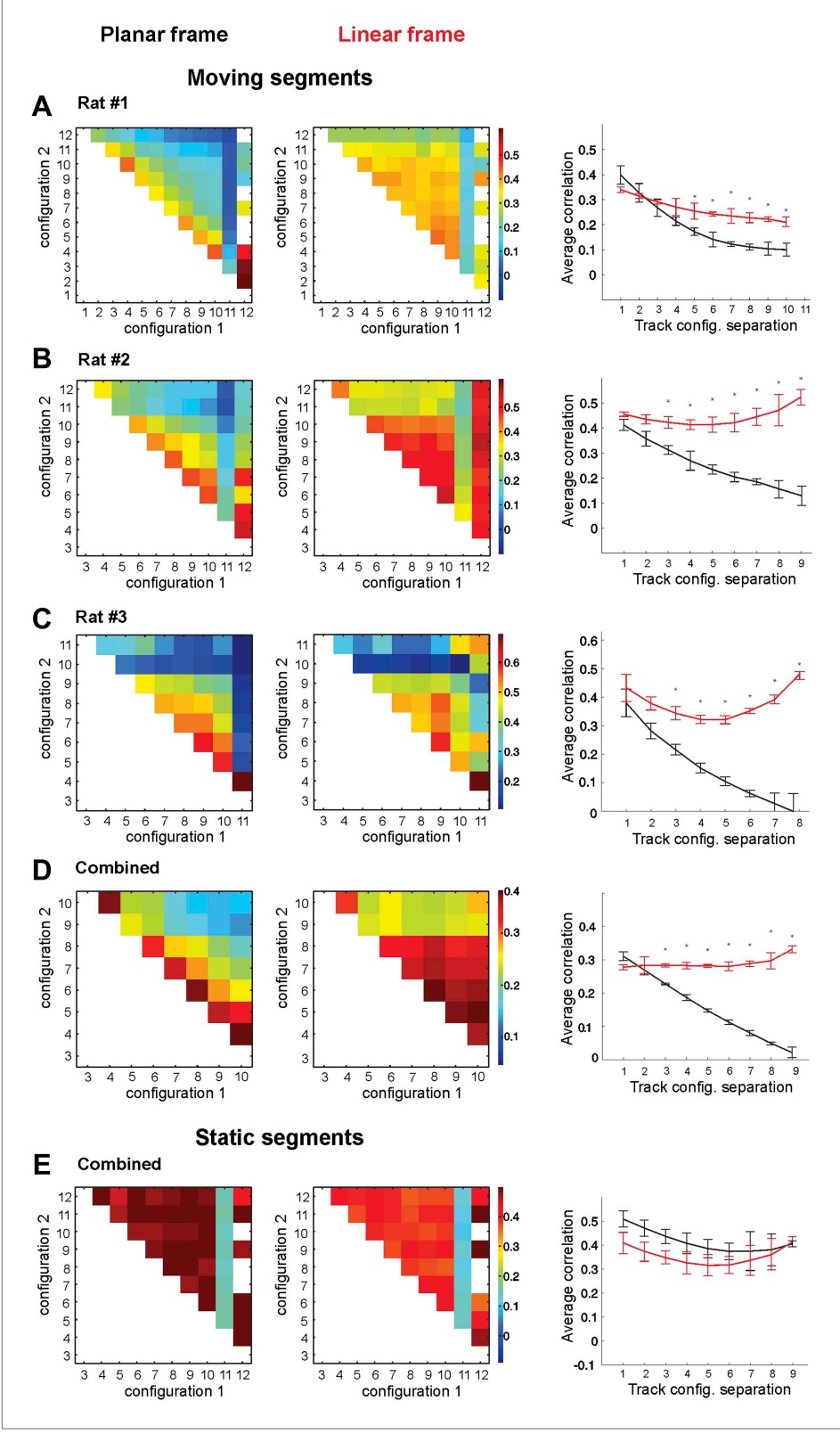

**Figure 3**. Place cell firing is stable across different track configurations. (**A**–**C**) The correlograms for the ensemble averages for each of three animals. The place cell spiking data are taken from only the segments of the track that change position between configurations. The two lines on the graphs in the third column represent the correlation

*Figure 3. Continued on next page*

*Figure 3. Continued*

coefficients of place cell ensembles in the 1D (red) and 2D (black) frames of reference. The correlation coefficients in the 1D configuration (red lines) remain stable, but the correlation coefficients for the 2D configuration (black lines) decay as the geometric difference between track configurations increases. (**D**) Combined data from all three rats for the correlation coefficients for the combined ensemble averages over all mobile track segments. Only data from the moving segments of the track and the corresponding correlations vs track configuration separation plot for the two reference frames are included. Firing across the population was much more stable in the linear reference frame. (**E**) The same plots as in **D** for spikes fired over the static sections of the track. Error bars represent SEM and * indicates p < 0.001.

animals on a U-shaped track that consisted of 11 segments, three of which could form a protrusion on either of the two sides (Track B, see *Figure 4A*, 'Materials and methods'). The geometric transformation in this case consisted of keeping the protrusion on one side of the track during the first session and then flipping it to the other side during the second session. The starting side for the protrusion alternated each day. On days 1, 3, and 5 during the first 20-min running session, the protrusion was formed by segments 3, 4 and 5; during the following 20 min running session, segments 7-8-9 formed the protrusion (*Figure 4B*). On days 2, 4 and 6 the protrusion first appeared in position 7-8-9 and then flipped to the opposite side for the second session (*Figure 4C,D*). In order to enter the protrusion area, the animal turned to the right on its inbound journeys and to the left on its outbound journeys during both runs, before and after the flip of the protrusion. The protrusion switch thus did not affect the sequence of turns but did produce a local deformation of space, which altered the relative arrangement of the track's segments and changed the distance relationships between locations on it. As before, the deformation did not violate the track's topological structure. Unlike the Track A experiment, however, the geometry of track B changed discontinuously, as the animal was physically taken off the track between the two running sessions.

The geometric transformation provided by Track B allowed us to conduct analyses that complemented our observations on Track A. First, we asked whether place cell activity was bound to the local features on the track (individual segments and joints). If so, place fields would tend to move from one side of the track to the other with the protrusion. Second, we could verify whether place cell activity

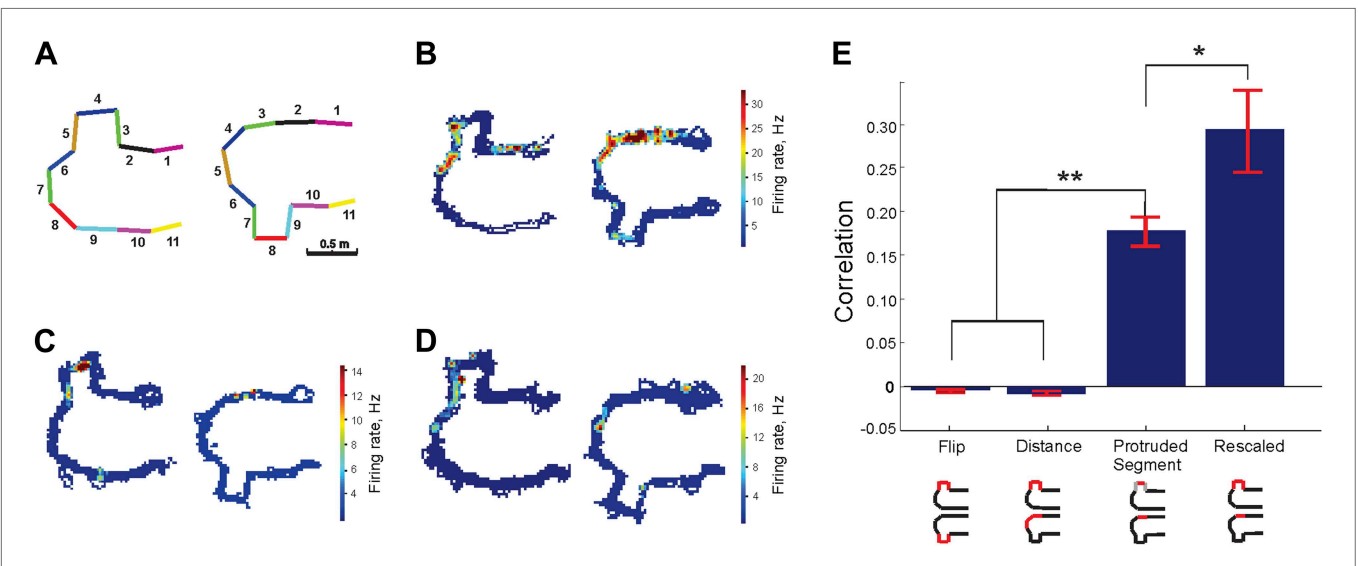

**Figure 4**. Protrusion-flip track (Track B). (**A**) Schematic representation of Track B, with segments numbered, showing the position of the track's protrusion during two successive run sessions. (**B–D**) Each show a representative place cell firing in the two configurations. (**E**) Correlations for each of the four comparisons of regions across the two configurations. The rescaled correlation is highest, indicating that the data are most compatible with a scheme in which the place cell firing does not simply respond to changes in geometry. Error bars represent SEM with * and ** indicating p < $10^{-8}$ and < $10^{-9}$, respectively.

was determined by the animal's linear distance along the track. If this were the case, a population of place cells would remain active at the same distances from the ends of the track before and after the protrusion switch, that is, they would shift along the track as a result of the protrusion's flip. In contrast, our topological model would predict that the place field in the region of the protrusion should not flip over or shift; instead, it should simply stretch and shrink as the protrusion is inserted and removed. So if place cell spiking activity represents connectivity, the place fields on the section of track replaced by the protrusion should extend continuously into the space provided by the protrusion.

We tested these hypotheses by recording from each animal for 7 days: a total of 114 putative excitatory pyramidal cells were recorded while animals traversed Track B. As shown in the examples in *Figure 4B–D*, individual place cells were generally not bound to the protrusion but instead tended to stretch flexibly along the track when the protrusion was moved. To quantify the changes induced by the movement of the protrusion on the population level, we computed the ensemble averages for several types of correlations between the firing rates on the two protruding regions before and after the protrusion flip. All of these correlations were computed using the linearized data to ensure fair comparisons and included all cells that had a peak occupancy normalized firing rate above a 2 Hz threshold. This low threshold was chosen to be as inclusive as possible of different place cell firing patterns.

In our analysis, we saw no tendency of CA1 place fields to remain at fixed distances from the ends of the second track (Track B). Our findings cannot be explained by place cells responding purely to local features (such as junctions) or measuring distances from those features, since the place fields showed no tendency to be bound to specific track sections on the protrusion. Instead, place fields stretched or shrank to reflect the insertion or removal of the protrusion. The place fields were best described as remaining in their relative locations on the track. The bar graph in *Figure 4E* shows the mean of the resulting values of the correlation coefficients for four cases.

First, we considered the possibility that the place fields remained bound to specific features of each segment of the track. The first bar, marked 'Flip', shows the average correlations of the firing rate over the protrusion across the two configurations. Since this value is negative and close to zero, it is clear that the place cells on average showed no tendency to be bound to individual segments of the protrusion. The second bar, 'Distance', shows the value of the firing rate correlations between the sections occupied by protrusion in the first configuration (sections 3–5 or 7–9, depending on which way the protrusion was facing) and the sections at the same distance from the wells from the second configuration. Similarly to the previous case, the correlations for the distance-based comparison were close to zero; moreover, the difference in the mean correlation values between cases 1 and 2 was not statistically significant. The third bar, 'Segment', shows the average firing rate correlations over the protruding section in the first configuration (i.e., sections 4 or 8) and the section onto which the center segment of protrusion projects directly after the flip (sections 3 and 9, respectively). Clearly this correlation is significantly higher than in the previous two cases. The low protrusion-to-protrusion 'Flip' activity correlation values compared to the protruding 'Segment' correlations demonstrate that most or all place cells that were active on the protrusion in Configuration I did not follow the protrusion when it was moved to the other side but instead remained in the same relative positions on the track.

Lastly, the bar 'Rescaled' shows the firing rate correlations following rescaling of the protrusion length (and the coordinates of the corresponding spikes) to compensate for the spatial deformation produced by protrusion move (e.g., scaling the total length of the sections 3, 4 and 5 down to the length of the section 3 that replaces the protrusion after the flip). Remarkably, the rescaled correlates were not only significantly larger than those of the local features (case 1) and distance (case 2) scenarios, but also were larger than the protruding segment correlations, indicating that local stretching could describe a greater amount of the variability in place fields than path integration (non-parametric $u$-tests, which give the p values $10^{-5}$ for the differences and one way ANOVA, Tukey–Kramer post-hoc tests, p's $< 10^{-4}$). The rescaled case illustrates that rate remapping compensated for the spatial expansion in such a way as to not alter the structural framework of the map.

The average correlations on Track B were relatively low compared to those on Track A. This could reflect firing variability because the 'flip' of the protrusion Track B happened discontinuously, in one step, leading to some degree of disruption of the continuity of the environment (similar to [*Diba and Buzsaki, 2008*], where the transformation also induced partial remapping). We hypothesize that, were we able to carry out this experiment using a continuous transformation from the straight section of the track to the protrusion, as we were able to do with Track A, the hippocampal network would have retained a greater degree of similarity across the configurations. Nevertheless, the difference in the

correlations between these four cases was statistically significant. The fact that both the mean and individual place cell activity across the two sessions showed the highest correlation after the rescaling transformation was applied, while the 'Distance' and 'Flip' correlations were low, suggests that place cell firing, despite its variability, served primarily to represent the connectivity of the environment through the temporal sequence of place cell activity. The CA1 spatial representation thus remained invariant with respect to a wide range of geometric transformations.

## Discussion

By explicitly counterposing geometry and topology in this series of experiments, we have demonstrated that hippocampal place cells show a remarkable degree of stability with respect to even dramatic geometrical changes in the environment: they retained their relative order and connectivity despite routes involving diametrically opposite protrusions and orientations. As long as the sequence of spaces experienced remains the same, in other words, and the place fields still overlap, it would appear not to matter to the hippocampus if the place fields are stretched by two centimeters or twenty. We therefore propose that place cells do not represent locations in space but rather provide a spatial context for experience (*Eichenbaum et al., 1999*; *Moser et al., 2008*).

This would seem to be more compatible with the higher-order functions of the hippocampus in representing and storing experience, and it is consistent with results from experiments that have been conducted in morphing environments, which have an implicit, if overlooked, topological aspect. *Muller and Kubie (1987)* found that enlarging their open field set-up caused place fields to enlarge while maintaining the same shape and relative order (i.e., the connectivity of the environment remained invariant). Similar observations were made by *Gothard et al., 1996a* on a linear track, by *O'Keefe and Burgess (1996)* in rectangular environments, and by other studies in morphing environments (*Lever et al., 2002*; *Leutgeb et al., 2005*; *Touretzky et al., 2005*; *Wills et al., 2005*; *Colgin et al., 2010*). This phenomenon is topological in character, since the fields stretch with the space but maintain their basic position and sequence. A more detailed analysis of place cell activity in a stretching *1D* environment reported in *Diba and Buzsaki (2008)* shows that the temporal relationships between the firing of the relevant place cells remains fixed throughout the stretch, despite the change in the overall firing rate (rate remapping, *Dupret et al., 2010*).

In addition to these previous studies showing that place fields stretch (maintain their sequence and relative positions) in morphing environments, we can add several other compelling reasons to doubt that the hippocampal map is exclusively or fundamentally Cartesian, like a street map. There is our recent modeling work showing the feasibility of spatial learning based on topological information (*Dabaghian et al., 2012*; *Arai et al., 2014*); early studies by Piaget and Inhelder showing that children first conceive of spatial relations in topological terms and only later think in terms of geometrical positions (*Dawson and Doddington, 1973*); and previous studies on rodent navigation showing that place cells respond to topological changes in the environment such as the placement or removal of a barrier across a previously learned route (*Poucet and Herrmann, 2001*; *Alvernhe et al., 2011*, *2012*). We propose that a topological framework, more like a subway map, would provide the hippocampus a more powerful, flexible, and readily formed spatial substrate for creating experiential memories or evaluating the spatial context of behaviors (*Lavenex and Amaral, 2000*; *Banquet et al., 2005*). Indeed, one study found that cells in the rat medial prefrontal cortex, recipients of place cell efferents, did not fire in response to the rat's position or head direction but rather in response to specific goal-oriented behaviors (*Poucet, 1997*).

All this is not to say that other brain regions do not contribute metrical information to the internal spatial map—there is abundant evidence that many regions are involved in providing sensory, idiothetic and proprioceptive information for path integration (*Knierim et al., 1996*, *1998*; *Fenton et al., 2000a*, *2000b*; *Quiroga et al., 2005*; *Yoganarasimha et al., 2006*; *Moser et al., 2008*)—only to propose that the hippocampus serves a more general role in spatial cognition by providing a qualitative, flexible representation of the environment. We suggest that place cells emphasize the connections between portions of a given environment and the sequence in which they are experienced. Such an encoded sequence is evident in spontaneous replays of place cell spiking during quiescent (*Muller and Kubie, 1989*; *Davidson et al., 2009*; *Karlsson and Frank, 2009*; *Carr et al., 2011*; *Dragoi and Tonegawa, 2011*; *Zeithamova et al., 2012*) and sleep states (*Skaggs and McNaughton, 1996*; *Louie and Wilson, 2001*; *Foster and Wilson, 2006*), which recapitulate the order in which the place cells fired during navigation. This implies that when the hippocampal network is driven internally, the

structure of sequences remains the same as during navigation, although the activity of place cells under these circumstances is clearly not coupled directly with spatial locations. The encoded map thus appears to serve as a template for generating replay sequences and facilitating imaginative navigation. More recently, studies of hippocampal preplay (as opposed to replay) indicate that the hippocampus has a repertoire of preconfigured temporal place cell firing sequences that can be called upon to rapidly encode multiple novel spatial experiences (*Dragoi and Tonegawa, 2013*). It is striking that studies from such diverse fields, from electrophysiology to algebraic topology, seem to converge on the notion that the hippocampus operates by providing a flexible, rough-and-ready framework into which spatially grounded experiences can be organized and remembered.

If our topological model is borne out, it will have several implications. It is well known that if the environment changes significantly, then place cell activity can undergo global remapping, and if the changes are small, then the place cells preserve the relative order of spiking and respond only by a regular change of their spiking rates (rate remapping) (*Colgin et al., 2008*; *Allen et al., 2012*). Our topological hypothesis predicts that, within limits (which are as yet unknown), geometric transformations should cause only rate remapping while the sequence of firing remains invariant. Our topological model would also greatly narrow the realm of meaningful correlations between changes in visual cues and responses in place cell spiking. As we have tried to emphasize in this work, a place cell response does not necessarily translate into information communicated downstream. In the absence of global remapping, it may thus not be safe to interpret correlations between changes in stimuli and place cell activity as representing information that is transmitted to other brain regions. A change in response would primarily serve to establish the scope of geometric invariance, which could depend on a variety of biological conditions and which is a question that must be left for future investigations. Moreover, since the sequence of spaces traversed is encoded in ensembles of place cells rather than individual neurons, it is place cell ensembles, not individual neurons, whose responses to changes in the environment need to be studied. This of course poses technical challenges, but such challenges are more likely to be met if we see the necessity of doing so.

The notion that the hippocampus encodes topological relationships could help clarify the spatial vs non-spatial debate that has gone on for many years (*Eichenbaum et al., 1999*; *O'Keefe and Nadel, 1978*). Indeed, it is well known that the rat hippocampus is important for learning and remembering various types of non-spatial information, including visual, odor and action sequences (*Agster et al., 2002*; *Fortin et al., 2002*, *2004*; *Sauvage et al., 2008*). These functions, and the importance of the human hippocampus for episodic memory (*Squire, 1992*), have led to the suggestion that the hippocampus performs a more general associative memory function (*Eichenbaum et al., 1999*; *Wood et al., 1999*). It is not entirely clear how the same mechanism would be responsible for representing geometry (distances and angles) in the spatial domain and encoding sequences and events in the mnemonic domain. The beauty of the topological hypothesis is that hippocampal spatial maps would have the same structure as memory maps or 'memory spaces' (*Eichenbaum et al., 1999*; *Eichenbaum, 2000*), which would not only explain the stability of relative firing order in flexible environments, but could also provide a common framework for understanding spatial representation, spatial memory and episodic memory mechanisms.

## Materials and methods

### Spatial environments

We built the tracks as follows: track A (*Figure 1A*) consisted of 10 straight segments 35–40 cm long each (total length ~4 m), shown in color in *Figure 1B*. The width of the segments (~8 cm) and the height of the rim (3 cm) were fixed over the whole length of the track with the exception of the circular joint areas (120 mm in diameter) between the track segments that had no outer rims. While the two leftmost segments, 5 and 6, remained fixed, the four segments 1–4 of the top arm and the segments 7–10 of the bottom arm could move, so that segments 1 and 10 could be displaced over the full span of ~140 cm each.

During the recordings, each arm was moved over a selected distance on four remotely controlled platforms positioned on a system of rails, visible in *Figure 2A*, driven independently by two stepper-motors. The distance moved varied pseudo-randomly across trials. The motions of the two arms were unrelated to each other: a contraction phase of one arm could be accompanied either by contraction or by expansion of the other arm, and the distance the arm moved varied pseudo-randomly across trials.

Our goal of studying the geometric invariance of the place cell maps imposed a number of constraints on the experimental design. For example, embedding a flexible environment into a stable background would unavoidably produce geometrically complex conflicts between the local and the distal cues (*Knierim et al., 1996*; *Shapiro et al., 1997*; *Knierim, 2002*; *Knierim and Rao, 2003*). To avoid such issues and to control for information about the track configuration that the rat could infer from changes in the relative positions of the objects in the room, we excluded distal cues by conducting the experiment with no external lighting. To facilitate the rat's navigation, we covered the track surfaces with a scentless, washable, nontoxic glow-in-the-dark paint (Mister Art ASTM D-4236) that provided a soft green glow but not enough light for the distal walls to be visible. The entire visible space was thus concentrated around the track. Furthermore, the view of one glowing arm of the track from the other was blocked by a large screen so that the rat could see only the arm of the track he was traversing. The stepper motors were relatively quiet, and the animals did not display any apparent reactions to the sounds or vibrations generated during the track position shifts.

During recordings on Track A, we moved the remotely controlled arms while the animal received the reward (Hershey's chocolate milk) at the food well on the tip of opposite arm, so as to avoid cuing the rat about the track configuration change. We adjusted the speed of the platform displacement so that the track arms could be fully expanded or contracted during the period in which the rat was consuming the reward while also minimizing track vibrations and motor noise. There were a total of 7 days of recording for each animal on track A that included 2 run sessions and 1 sleep session. During each run session, both arms reached the maximally contracted and the maximally stretched positions a few times. The exact number of track moves on each day depended on the animal's foraging activity and ranged between 10 and 80 moves, with an average of 40 moves.

Track B (*Figure 4A*), used for control recordings in CA1 (see below), contained 11 segments made with the same material as the Track A, with a total length of 4.5 m. During each running session three segments formed a protrusion on one or the other side of the track. Two animals were exposed to Track B. For one of these animals the exposures were interleaved with exposures to Track A (with 10 hr' separation between exposures to track A and track B), while the other animal experienced only Track B. Each animal had 7 days of experience on Track B. Each recording session consisted of two 20 min runs and two 15 min sleep sessions in a run-sleep-run-sleep sequence. On days 1, 3 and 5 the protrusion was formed by segments 3, 4 and 5 during the first run session; during the first sleep session the same physical sections were 'flipped' to the opposite side, to positions 7-8-9 (*Figure 4B*). On days 2, 4 and 6 the protrusion was first formed by sections 7-8-9, then flipped to the 5-4-3 position (*Figure 4C,D*).

## Behavioral training

All the experimental procedures were approved by the Institutional Animal Care and Use Committee at UCSF. A total of four male Long-Evans rats were used in two separate groups of experiments. During the 2 weeks prior to surgery, the animals were pre-trained on a static U-shaped track to alternate between two food wells located at the ends of the track for chocolate milk rewards. During pre-training and after recovery from surgery, the rats were kept at 85% of the free-feeding baseline weight. Outside of the experimental recording periods, the rats had unlimited access to water in their home cages and were kept on a 12:12 hr light–dark cycle. The experimental testing took place during the dark phase of the cycle. At the end of the training period the rats were implanted with a 30-tetrode drive (see *Karlsson and Frank, 2008* for details). About half of the tetrodes were lowered directly into the CA1 area of the hippocampus and the other half remained in the parietal cortex, located above the CA1 area (cannulas at (−4, ±2.1) and (−4.8, ±4.4)).

During the recordings, the animals had no difficulty adapting to the changing geometry of the track and showed no signs of being distracted by the slight noise of the motors or track vibrations. Since the pre-training track was fully stretched and static, the animals showed exploratory behavior on the flexible track during the initial recording sessions, by balancing over the edges and exploring the space surrounding the track segments, especially in the vicinity of the joints. Nonetheless, animals performed the task very well, completing the full trajectory from one food well to the other on 98% of runs. The average time required to complete the track (~34 s) did not differ significantly from the level of the animals' performance on the static track during pre-training. Given this reliable alternation behavior we were able to quantify the place cell responses on the track.

## Data analysis

We separated putative excitatory cells from putative inhibitory cells using the standard criteria based on firing rate, spike width and the mean ISI. All analyses excluded times when animals were moving on the track at a speed of less than 3 cm/s. Data analysis was performed using MATLAB.

Since we were interested in comparing place cell firing across different track configurations, we needed both a system to describe the track configurations (and allow comparison between different configurations) and a way of linearizing the track to allow us to compare firing in the 2-dimensional planar and 1-dimensional linear frames of reference. We used standard methods for binning the space to define precisely where the place cells fired. We describe our methods for each of these needs below.

## Defining different track configurations by motion of segments through different vertical sections of the 2D track space

The total number of track moves per day was large (up to 80 moves) and the rat spent only a short period of time in each configuration (about 34 s including the slow exploratory movements of the animal around certain segment junctions). In order to make the statistical analysis more robust, we combined similar track configurations in groups. Since the arms contract uniformly (i.e., the angles between segments 1 and 2, 3 and 4, etc., remain similar), the configuration of each arm can be specified by the position of the endpoints of the track (the food wells, F1 and F2). Therefore, it is possible to split the full area swept by the moving sections of Track A into a number of vertical sections and then to specify geometrically similar track configurations by the index of vertical sections occupied by the track's ends (*Figure 1D*). We used $N_{tot}$ =12 vertical sections on each arm of the track, 9 cm wide, see *Figure 1C* (note that the bottom sections are slightly displaced; the bottom arm of the track did not stretch fully because of a mechanical difficulty). The number $N_V$ of track configurations that fall into a particular section $V$ varied daily depending on the level of the animal's foraging activity. However, each vertical section **V** was populated by at least three track configurations, $N_V \geq 3$, during each day of recordings, that is, only configurations that were experienced at least 3 times were included in our analyses. This method of defining the track configurations, illustrated in *Figure 1D*, provides the basis for comparison of different track configurations that result in the matrices depicted in *Figures 2 and 3*.

## Linearization of the track

In order to directly compare the spiking activity in the planar and in the linear frames of reference, we used the planar coordinates (the usual *x-y* coordinates of the *2D* plane) and then linearized the coordinates for 1*D* such that $x_L$ provides the distance from one of the food wells, and $y_L$ provides the displacement from the midline of the track (recall that the track is 9 cm wide). As a result, the same area of physical space can be described in both the *2D* and the *1D* representations, which permits a fair comparison of correlations across the two reference frames.

## Binning the space and calculating correlation coefficients

We then binned the space into 3 × 3 cm squares in both the linear and in the planar frames of reference and computed the occupancy-normalized spatial rates for each place cell for all the track configurations that fall into a particular spatial bin (*Figure 1—figure supplement 1*). Since the global placement of the grid of spatial bins is arbitrary, we averaged the rates over the values obtained by shifting the grid randomly by up to a half-bin width in both the *x* and *y* directions. As a result, the firing rates for all cells in the planar and the linear reference frames are roughly the same and can thus be used as characteristics of a given place cell's activity both in the planar and in the linear perspective. No other smoothing was applied to the data. To ensure meaningful comparisons of the correlations between spatial firing patterns in linear and planar reference frames, we considered only those bins (1) that were actually visited by the rat in both configurations and (2) in which at least one spike was fired.

Since the animals spent only a relatively short time on each segment of the track during each run, an activated place cell typically produced only a few spikes. Given that place cells are highly variable (*Brown et al., 1998*; *Fenton and Muller, 1998*), we combined multiple passes through a given place field in similar track geometries in order to obtain meaningful results ('Materials and methods' and *Figure 3D,E*). This reduced the variability of our estimates of the rate while preserving distinct estimates for different shapes. Nonetheless, our estimates were based on relatively few (3 to 7) passes through each location, which lowers the correlations of place cell activity rates on the moving segments of the track, as compared

to the stable segments (5 and 6). For this reason, we analyzed the spiking occurring on the mobile sections of the track (sections 1–4 and 7–10) separately from the spiking at fixed regions (sections 5–6).

## Acknowledgements

We thank Daoyun Ji, M Carr, S Cheng, M Karlsson, C Kemere, S Kim, A Nathe, and A Singer for thoughtful comments on the manuscript. This work was supported by the Swartz and Sloan Foundations, NIH grants NIH 5F32NS054425 and MH080283, the W. M. Keck Foundation, Houston Bioinformatics Endowment Fund (YD).

## Additional information

### Funding

| Funder | Grant reference number | Author |
| --- | --- | --- |
| National Institute of Neurological Disorders and Stroke | 5F32NS054425 | Yuri Dabaghian |
| National Institutes of Health | MH080283 | Loren M Frank |
| W.M. Keck Foundation | Houston Bioinformatics Endowment Fund | Yuri Dabaghian |

The funders had no role in study design, data collection and interpretation, or the decision to submit the work for publication.

### Author contributions

YD, Conceived of the idea, Conducted the experiments, Analyzed and interpreted data, Wrote and edited the manuscript; VLB, interpreted data, wrote and edited the manuscript; LMF, Acquisition of data, Analyzed and interpreted data

### Ethics

Animal experimentation: All the experimental procedures were approved by the Institutional Animal Care and Use Committee at UCSF, Approval Number: AN081431-03D

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
