## [Decision Letter]

Thank you for sending your work entitled “Reconceiving the hippocampal map as a topological template” for consideration at *eLife.* Your article has been favorably evaluated by Eve Marder (Senior editor) and 3 reviewers, one of whom is a member of our Board of Reviewing Editors.

The Reviewing editor and the other reviewers discussed their comments before we reached this decision, and the Reviewing editor has assembled the following comments to help you prepare a revised submission.

This study offers the strongest evidence to date that hippocampal networks represent spatial contiguities and not 2D Cartesian maps created by path integration. The experiment directly pitted against one another these two hypotheses by having rats run in a variable shaped track in the near-dark, and CA1 place cells tracked the linear contiguities much better than the 2D shapes created by various configurations. The following comments are intended to improve the manuscript and better support the authors’ central claim.

1) The second full paragraph of the manuscript describes how maze segments and bins are used to measure geometric similarity and refers to Figure 1, which includes a panel D that is not described in the figure legend or text. The track bins shown in the panel include only segments 1-3 and 9-10, and though all of the segments should be binned similarly, it seems, the caption describes a 12 bin configuration. Please clarify. An example illustrating exactly how the bin indices and track configurations are used to define track similarities and firing correlations would be helpful.

2) The paper emphasizes the improved prediction of spike activity by considering topology, “... so that the flexible arrangement of place fields does not represent specific locations on a Cartesian grid, but rather a coarser framework of spatiotemporal relationships.” While the evidence does show that sequential representations are more reliable then topographic ones, the experiment did not test topological mapping per se, e.g. by recording CA1 activity in topologically different tasks. Without these kinds of data, the “topological template” and other descriptions of hippocampal function become quite similar, e.g. to Eichenbaum's “memory map” and Worden's “fragment fitting” ideas.

3) The examples shown in Figure 1 and E,F are a tad confusing, unnecessarily. We assume that the linear representation has the upper right end of the “U” at the left. This could be easier to understand by indicating the corresponding origins of each representation and perhaps also by putting the linear origin on the right for better correspondence.

4) The key analysis in this paper is a correlation analysis but the analysis is poorly described and it is not at all clear what was done. One can imagine the matrices in Figures 2 and 3 represent the average correlations between a single place cell's activity when the maze was in two distinct configurations indicated as bin 1 and bin2, but this is not clearly communicated. The authors should make a diagram in the Methods to explain what is being compared.

5) The authors rely on the correlation analysis (that is not transparent) to make their point. They find that place cell firing across the maze configurations is more consistent (higher correlation) if the end of the arm is the origin of the coordinate system than if a standard Euclidean system is used. It seems that the essence of the central claim is that place cell discharge encodes topological rather than geometric variables. If this is the case then the authors should be able to better decode the topological variables than the geometric variables. An appropriate comparative decoding analysis should be done to evaluate this core assumption.

6) Another main concern is that in earlier work such as Gothard et al. (J Neurosci 1996a), and Redish et al., (J Neurosci 2000), experimentalists have done similar manipulations and recordings to the current work's. The prior results were interpreted in terms of the hippocampal coding of distinct geometric spatial reference frames. From a topological point of view it is not obvious why the lengthening/shrinking of the linear environment as in previous work, is an important difference from the kinking of the environment in the present environment. Consider also the work with hairpin mazes such as Derdikman et al (2009) and open fields such as Gothard et al (J Neurosci 1996b). Importantly, in previous work it was found that place cells did not represent geometric space in a simple manner. The prior results resemble the data in Figure 1. These observations were interpreted as the rat representing space in different spatial frames, where the origin of one frame is the initiation of a run and the origin of the other frame is the end of the run, but both being geometric representations. Why is this not a parsimonious and equally accurate explanation for the present data?

7) The authors should address the possibility that the hippocampus does represent 2D geometry when the geometry is reliable (as in most stable open field environments), and that the better tracking of linear contiguity here is due to the design by which contiguities were reliable and the 2D environment was not. Note that, in previous cases where hippocampal neurons shrink or expand to distortions of the environment, systematic variations in those distortions were a major parameter of the task design, whereas more categorical responses (re-mapping) is observed when environments and task parameters are discontinuously changed (as in their track B).

Minor comment: For the sake of non-mathematicians, please define topology as opposed to geometry.

---

## [Author Response]

For the sake of non-mathematicians, please define topology as opposed to geometry.

We define the topology of the environment as a set of qualitative relationships between its parts: adjacency, containment, connectedness, overlap, cover, etc. (2; 68). We now include this more explicit definition in the Introduction. These qualitative relations stand in contrast to metrical (geometric), quantitative features such as distances, angles, and shapes. An analogy that makes the distinction easier to remember is that a geometric or Cartesian coordinate map is akin to a to-scale street map, whereas a topological map is akin to a subway map, which doesn’t have to be to scale to be useful; it simply has to tell you which lines connect with which at which stations, how many stops there are until you need to get out, and so forth. It can be very much not to scale but still be perfectly functional. Geometry is much less forgiving of approximation, as the frustrations of inaccurate street maps make perfectly clear.

It is crucial to note that all space is characterized by both qualitative and quantitative (topological and geometric) features, but the relation between these aspects of space is asymmetrical: qualitative spatial relationships are easy to deduce if the identity, shapes, and sizes of specific regions in the environment are known. The converse is not true: topological relationships do not define the shape or scale of either the parts that they apply to or of the environment as a whole. For example, the fact that region *x* is adjacent to or overlaps with the region *y* says nothing about the shapes or sizes of *x* or *y*. The latter are defined via geometric measures (distances, areas, angles, etc.), and the relationships that are built on them.

*This study offers the strongest evidence to date that hippocampal networks represent spatial contiguities and not 2D Cartesian maps created by path integration. The experiment directly pitted against one another these two hypotheses by having rats run in a variable shaped track in the near-dark, and CA1 place cells tracked the linear contiguities much better than the 2D shapes created by various configurations. The following comments are intended to improve the manuscript and better support the authors’ central claim*.

*1) The second full paragraph of the manuscript describes how maze segments and bins are used to measure geometric similarity and refers to*
Figure 1*, which includes a panel D that is not described in the figure legend or text. The track bins shown in the panel include only segments 1-3 and 9-10, and though all of the segments should be binned similarly, it seems, the caption describes a 12 bin configuration. Please clarify. An example illustrating exactly how the bin indices and track configurations are used to define track similarities and firing correlations would be helpful*.

We regret the lack of clarity; we have revised Figure 1 and its legend to give a much more detailed description of the binning and our approach to defining track configurations and correlations. In the course of doing so we realized we had numbered the planar track going from right to left (or rather, counterclockwise, because the opening of the U faces the right), but the linear track going from left to right; we have now flipped the linear tracks in both Figures 1 and 2 so that the segments are numbered from right to left, to more clearly parallel the U-shaped track.

*2) The paper emphasizes the improved prediction of spike activity by considering topology, “... so that the flexible arrangement of place fields does not represent specific locations on a Cartesian grid, but rather a coarser framework of spatiotemporal relationships.” While the evidence does show that sequential representations are more reliable then topographic ones, the experiment did not test topological mapping per se, e.g. by recording CA1 activity in topologically different tasks. Without these kinds of data, the “topological template” and other descriptions of hippocampal function become quite similar, e.g. to Eichenbaum's “memory map” and Worden's “fragment fitting” ideas*.

There are actually three aspects to this comment, all very important.

First, testing “topological mapping per se”: as we mention in the Introduction and Discussion, the Poucet lab in France has, in several very elegant and cleverly designed studies, recorded CA1 activity in topologically different tasks. For example, (5) placed rats in complex environments in which multiple doors and entries allow movement in and out of different sectors of the space, and they changed the topological structure (connectivity) of the environment by closing and opening different passages. Some doors did not alter the topology of the environment (did not alter the ability to move in between sectors) while others did, and the rats’ place cells clearly responded to changes in the connectivity between sectors of the space and not merely the manipulation of the doors. These results support our contention that the hippocampus is interested in topological features of space. They also accord with Piaget’s early studies of children, which demonstrated that young children first develop a sense of spatial relations that we would call topological and only later become able to conceive of space in geometric terms.

That said, and this is the second point, we are not particularly interested in whether place cells respond to topological features per se, nor indeed whether place cells respond to any particular cue (of which there are many); we are ultimately interested in what sorts of information hippocampal place cells are able to transmit to the rest of the brain. It is important to remember that the only information available to downstream regions must be contained in the temporal pattern of place cell firing. We believe that one of the unwitting limitations of place cell research has been the failure to fully appreciate the difference between the information we as neuroscientists are able map experimentally and that which is available within the subject’s brain. Unfortunately, we don’t know the identity of these downstream neurons and are thus forced to be imaginative (Buszaki has made this point beautifully in several papers).

Third, we agree that the “topological template” leads to a result very similar to Eichenbaum’s memory map (referred to in the manuscript) and Worden’s fragment fitting ideas (Worden, 1992) which we take to be a good thing. That three vastly different approaches converge onto very similar conclusions would seem to be an important validation of our topological hypothesis, which arose from a completely different starting point. In fact, as we were preparing these comments we came across a paper by Dragoi and Tonegawa (22) on hippocampal pre-play (as opposed to replay) that also converges onto similar conclusions: the paper suggests that the hippocampus has a repertoire of preconfigured temporal place cell firing sequences that can be called upon to rapidly encode multiple novel spatial experiences. We view such temporal sequence templates as a manifestation of the hippocampus’ need for rough, rapid conceptualization of spatial environments for purposes of navigating to feeding sites, nests, foraging, etc., in environments that can change appearance a great deal over the course of days and seasons (and would thus be not only computationally costly but wasteful to map out in precise geometric detail according to visual cues that could be covered in snow or erupt in riotous bloom overnight).

*3) The examples shown in*
Figure 1
*and E,F are a tad confusing, unnecessarily. We assume that the linear representation has the upper right end of the “U” at the left. This could be easier to understand by indicating the corresponding origins of each representation and perhaps also by putting the linear origin on the right for better correspondence*.

We appreciate the reviewers calling our attention to the ambiguity: by following the numbering scheme in the original Figure 1, we unwittingly committed ourselves to a counterintuitive presentation. We have revised Figures 1 and 2 so that the planar and linear frames of reference run in the same direction, with F1 at the right in both planar and linear frames.

*4) The key analysis in this paper is a correlation analysis but the analysis is poorly described and it is not at all clear what was done. One can imagine the matrices in*
Figures 2 and 3
*represent the average correlations between a single place cell's activity when the maze was in two distinct configurations indicated as bin 1 and bin2, but this is not clearly communicated. The authors should make a diagram in the Methods to explain what is being compared*.

The reviewers’ understanding is precisely correct, and we have included a new figure (Figure 1—figure supplement 1) in the Methods to better explain how we calculate the correlation coefficients.

*5) The authors rely on the correlation analysis (that is not transparent) to make their point. They find that place cell firing across the maze configurations is more consistent (higher correlation) if the end of the arm is the origin of the coordinate system than if a standard Euclidean system is used. It seems that the essence of the central claim is that place cell discharge encodes topological rather than geometric variables. If this is the case then the authors should be able to better decode the topological variables than the geometric variables. An appropriate comparative decoding analysis should be done to evaluate this core assumption*.

There is an important clarification to be made here: the reviewers are thinking in terms of coordinate systems, with the difference between our and prior experiments simply being the point of origin, but this is not the case. First, our experiments in Track B do address whether the starting point makes a difference in our results (it does not; see manuscript, and Figure 4). More importantly, however, in our model, there is no coordinate system, at least for hippocampal place cells (as we describe in the discussion, this is most definitely not to say the brain does not fill in geometric/coordinate information, simply that we are proposing that the hippocampus has a higher-order function in spatial cognition).

In addition, there are no topological variables here, since the topology is held constant. Topological variables would be whether particular places are connected/overlap, which can be detected via place cell coactivity. In contrast, the geometric variables here are the shifting angles that define the track’s various configurations. Our analysis implies that the place cell code is shape-invariant, which means that the geometric variables are not encoded at all.

We want to emphasize here a key concept that we introduce in the Discussion, which is useful to bear in mind as we respond to the reviewers’ additional points below. Our experiments reveal that hippocampal place cells retain their relative order and connectivity despite dramatic and diametrically opposite protrusions, orientations, and idiothetic input. We therefore propose that place cells do not represent abstract locations in allothetic space per se but rather provide a spatial context for experience. This makes sense in that the rodent must actually physically move through the place field (it is not something that the animal projects onto the experimental space just from visually surveying it or mapping it onto an imaginary Cartesian grid). This is in line with Eichenbaum’s memory space hypothesis, with hippocampal preplay, and with psychological studies demonstrating the importance of space and spatial metaphors in grounding memory and cognition.

*6) Another main concern is that in earlier work such as Gothard et al. (J Neurosci 1996a), and Redish et al., (J Neurosci 2000), experimentalists have done similar manipulations and recordings to the current work's. The prior results were interpreted in terms of the hippocampal coding of distinct geometric spatial reference frames*.

The results of prior studies in typical morphing environments ([16]; Gothard et al., 1996; [51]; [52]; [62]; [65]; Redish et al., 2000; [79]; [80]) are perfectly compatible with a topological perspective: enlarging the environment causes place fields to stretch elastically while maintaining their relative placement and sequence with respect to each other. In fact, we discuss some of these experiments in the Introduction in order to point out that their results were early, if inadvertent, indicators that place cells can behave in a topological manner.

*From a topological point of view it is not obvious why the lengthening/shrinking of the linear environment as in previous work, is an important difference from the kinking of the environment in the present environment*.

There is no difference from a topological point of view: the ‘kinking’ of the environment and the movement of the track arms in our set-up are, rather, designed to introduce a much more radical geometric variability in order to give the best possible chance for changes in angle of motion and distance traveled (relative to the underlying spatial grid) to evoke a response in place cells if indeed they serve to transmit such geometric information to downstream neurons. As illustrated in Figure 5, it is not at all clear that expanding the environment (and stretching place fields) would be transmitted downstream immediately as geometric information; it would undoubtedly require ensemble coding, but discussion of this topic is beyond the scope of the present concern. Suffice it to say that a few place cells are unlikely to transmit relatively subtle geometric changes downstream.Author response image 1.Metrical information contained in stretching place fields will not necessarily be conveyed by place cell spiking to downstream neurons. A) Three place fields (red, green and blue) stretch following the enlargement of the enclosure. To the right we show how the overlap between the place fields reflects the pattern of temporal overlap between the place cell’s spike trains; the rate of coactivity events depends on the distance between the place field centers, Δ_ij_ = r_i_ − r_j_, where ri is the distance of the i^th^ place field from the track’s center, their widths and the spiking rate amplitudes. B) After the environment expands over the distance D, the place fields’ layout stretches and the fields’ centers shift proportionally to their distances from the walls of the enclosure ([20]; Gothard et al., 1996). According to (65), some place fields may even remain at a fixed distance from the walls of the enclosure. In either case, the change in the distance between the place field centers is small. For example, consider two place fields on a 1-meter-long track located at r_1_ = 50 cm and r_2_ =60 cm, respectively. After the track stretches over D =100 cm, the place fields will move to r′_1_ = 100 cm and r′_2_ = 120 cm. As a result, if the original distance between place fields was Δ_12_ = 10 cm, after the stretch it will be Δ′_12_ =20 cm. The increase in distance between place field centers is thus only ε_12_ =10 cm, or 10% of the full stretch. Since place cell activity is modulated by an animal’s speed, visual cues, odor, and a plethora of other factors, a particular readout neuron would have no way of determining whether the change in firing rate is due to a geometric determinant, the animal slowing down, a change in a visual cue, or something else. Thus, even if the place field layout stretches significantly, the corresponding geometrical information would not necessarily be captured by the downstream networks unless the change is quite large.

*Consider also the work with hairpin mazes such as Derdikman et al (2009) and open fields such as Gothard et al (J Neurosci 1996b). Importantly, in previous work it was found that place cells did not represent geometric space in a simple manner. The prior results resemble the data in*
Figure 2
*and E,F. These observations were interpreted as the rat representing space in different spatial frames, where the origin of one frame is the initiation of a run and the origin of the other frame is the end of the run, but both being geometric representations*. *Why is this not a parsimonious and equally accurate explanation for the present data?*

The reviewers are correct that place cells in these studies did not represent geometric space in a straightforward manner; we would argue that this and the plethora of cues that apparently elicit place cell firing, among other considerations, are good reasons to view the purpose of place cells as something other than location-receptors. The apparent complexities of place cell behavior anchored in different frames of reference in previous experiments come at least in part from the seeming lack of awareness that there are any qualities of space other than geometric ones. In other words, many authors have never considered a topological alternative, so their interpretations of their data do not argue for geometry vs. topology, but simply assume geometry and interpret accordingly. The language used to describe certain results is thus very confusing if one actually realizes there are at least these two aspects of space. In the Gothard et al. paper (J Neurosci 1996b), for example, the Discussion has this passage: “Thus, the metric for the location representation appears not to derive from the perception of spatial relationships, such as the geometry of the environment, visual angles, and retinal image sizes of landmarks, but rather from self-motion cues.” From our perspective, it is misleading to equate “spatial relationships” with the geometry of the environment; we would say there are quantitative (geometric) and qualitative (topological) relationships. With that caveat, however, and within the framework of their assumptions, their conclusion is completely reasonable and consistent with our findings: the animals are not representing location through abstract visual representations of the geometry of the environment but from what Gothard et al. interpret as “self-motion cues” and what we interpret as a topological experiential framework.

Similarly, Derdikman and others (Derdikman et al., 2009; Singer et al., 2010) have shown that place cell activity is similar in geometrically similar locations. This also does not contradict our approach, because within each fragment the coding would be topological. We would predict that if the multiple-U maze used in these studies were to be continuously deformed so that its symmetry is preserved, then the place cell activity would remain invariant with respect to this deformation and the fragmentation would be preserved.

We believe it is far more parsimonious to view place cells as associating locations with particular experiences. It is not that the interpretations of previous experiments are not workable, but rather that a simple topological framework for experienced sequences of places that can be recalled in sleep or quickly called upon to fit novel environments [see (22)] seems not only more elegant but more consistent with other higher-order functions that have been established for the hippocampus. Furthermore, (1) spatial information is encoded by an ensemble of place cells rather than individual cells ([31]; Pouget et al., 2000), yet many experiments tend to treat place cells as individual feature detectors; (2) as noted in the manuscript and further explored below, there is reason to doubt that place cells would convey subtle geometric information to downstream neurons, and (3) there is the elegant pragmatism of quickly ascertaining the connectivity between aspects of our environment before requiring the enormous computational effort to make detailed calculations that would be necessary for a Cartesian coordinate map.

Last but by no means least, the topological hypothesis has allowed us to provide a mechanism to explain how place cells might be functioning as an ensemble to encode topological information [see (6; 17)]. This is an important consideration, as at this point path integration remains a black box.

*7) The authors should address the possibility that the hippocampus does represent 2D geometry when the geometry is reliable (as in most stable open field environments), and that the better tracking of linear contiguity here is due to the design by which contiguities were reliable and the 2D environment was not. Note that, in previous cases where hippocampal neurons shrink or expand to distortions of the environment, systematic variations in those distortions were a major parameter of the task design, whereas more categorical responses (re-mapping) is observed when environments and task parameters are discontinuously changed (as in their track B)*.

This gets to the heart of why we took the approach we did. Recall our explanation of the relation between geometry and topology at the very beginning of this rebuttal: topology can be inferred from geometry, but geometry cannot be inferred from topology. Thus, if the geometry of the environment is fixed, the topology is also fixed, so that if cells *do* respond to geometric determinants of the environment, this information can be used to produce topological characteristics of the environment. This would make it impossible to delineate, in stable environments, whether the cells are responding to topology or to geometry. Answering this question requires weakening the connection between geometry and topology, which we have sought to do here in our experiments.

In Author response image 1 we take a more in-depth look at why place cell firing patterns may not transmit geometric changes in an environment to downstream neurons. We could incorporate this figure into the manuscript if the reviewers and editors think it is useful.